# Decline and Extinction of the Italian Agile Frog *Rana latastei* from Core Areas of Its Range

**DOI:** 10.3390/ani13203187

**Published:** 2023-10-12

**Authors:** Gentile Francesco Ficetola, Raoul Manenti, Elia Lo Parrino, Martina Muraro, Benedetta Barzaghi, Valeria Messina, Simone Giachello, Andrea Melotto, Mattia Falaschi

**Affiliations:** 1Department of Environmental Science and Policy, Università degli Studi di Milano, Via Celoria 10, 20133 Milan, Italy; raoul.manenti@unimi.it (R.M.); elia.loparrino@uimi.it (E.L.P.); marti.muraro93@gmail.com (M.M.); benedetta.barzaghi@unimi.it (B.B.); valeria.messina@unimi.it (V.M.); simgia.sg@gmail.com (S.G.); andrea.melotto@unimi.it (A.M.); mattia.falaschi@unimi.it (M.F.); 2University Grenoble Alpes, University Savoie Mont Blanc, CNRS, LECA, Laboratoire d’Écologie Alpine, F-38000 Grenoble, France; 3Department of Biological, Geological and Environmental Sciences, Università di Bologna, Via Irnerio, 42, 40126 Bologna, Italy; 4University School for Advanced Studies IUSS Pavia, Piazza della Vittoria 15, 27100 Pavia, Italy

**Keywords:** global amphibian decline, climate change, drought, invasive alien species, *Procambarus clarkii*, EU habitat directive, population trends, amphibian conservation

## Abstract

**Simple Summary:**

The Habitats Directive is the main legislative tool to protect European biodiversity and aims to ensure the persistence of species listed in its annexes. Here, we document the changes in abundance and occurrence of the Italian agile frog (*Rana latastei*) within Monza Park, which currently represents the area closest to the type locality of the species. We found that the range of the species inside the park strongly contracted in the early 2000s due to the destruction of breeding sites. Then, around 2021, the species completely disappeared from the area, probably due to the joint effects of inappropriate management and droughts. The extant legal protection did not prevent the local extinction of the Italian agile frog from a core area of its range; hence, we call for more effective tools for the conservation of European biodiversity.

**Abstract:**

Detecting the trends of species and populations is fundamental to identifying taxa with high conservation priority. Unfortunately, long-term monitoring programs are challenging and often lacking. The Italian agile frog *Rana latastei* is endemic to Northern Italy and adjacent countries, is considered vulnerable by the IUCN, and is protected at the European level. However, quantitative estimates of its decline are extremely scarce. In this study, we document the trends in abundance and distribution of *Rana latastei* within Monza Park, which currently represents the area closer to the type locality of the species and holds unique genetic features. Wetlands within the park were monitored from 2000 to 2023; counts of egg clutches were taken as a measure of reproductive output and the abundance of breeding females. In 2000, the species occurred over a significant proportion of the park. Total abundance showed strong yearly variation but remained rather constant from 2000 to 2019. However, *Rana latastei* disappeared from the park around 2021 and was never detected in 2022–2023. The decline is probably related to the joint effect of multiple factors, including the conversion of breeding sites for farming, inappropriate water management, invasive alien species, and severe drought. The local extinction of *Rana latastei* occurred despite legal protection, highlighting the need for more effective and stringent tools for the conservation of European biodiversity.

## 1. Introduction

The dramatic global decline of amphibians was detected more than 30 years ago [1,2,3]. Amphibians are consistently indicated as the vertebrate group suffering the strongest declines and with the highest proportion of threatened species [4]. The latest IUCN report estimated that 41% of amphibian species are threatened with extinction, compared to 27% of mammals, 21% of reptiles, and 13% of birds (https://www.iucnredlist.org/resources/summary-statistics, accessed on 31 July 2023). Despite multiple calls and a growing number of ongoing conservation actions, conservationists have been so far unable to halt the global amphibian crisis.

Documenting amphibian declines and extinction is pivotal to identifying drivers and conservation priorities. However, ascertaining declines and extinctions is difficult, as they can be confused with natural fluctuations of populations [5,6]. For instance, the Costa Rican golden toad, *Incilius periglenes*, was last observed in 1989. Nevertheless, uncertainty persisted regarding the rate and timing of the observed decline, and several years were required to understand that its extinction was most likely determined by complex interactions between climate change and other stressors [7,8,9,10]. In 1996, 8 years after the last sighting and despite huge monitoring efforts, the golden toad was still considered “critically endangered” by the IUCN. Only in 2004, the golden toad was officially listed as “extinct” [11], after 15 years of regular quantitative monitoring without any detection [10]. This highlights how, in the absence of monitoring programs, a species can go extinct without a clear understanding of what may have led to the decline and without effective attempts to halt/reverse it. Regular, long-term monitoring is thus necessary to document demographic trends and propose management actions. Unfortunately, implementing and maintaining extensive and long-term monitoring is often challenging, making these actions jeopardized and rarely covering a significant portion of the species range.

The Italian agile frog *Rana latastei* is endemic to Northern Italy and adjacent countries and is classified as “vulnerable” by the IUCN because it is threatened by the joint effect of habitat loss and fragmentation, pollution, and interactions with invasive predators [12,13]. The type locality of the Italian agile frog is “around Milan” in Northern Italy [14], but this area is currently one of the most heavily human-dominated regions of Europe, with very few remaining natural habitats and wetlands potentially suitable for amphibians. The Monza Park (southern portion of the Lambro Valley Regional Park; Figure 1) is the area hosting the *R. latastei* population that is closest to the type locality of the species (i.e., Milan). Additionally, Italian agile frogs inhabiting this park are genetically distinct from conspecific populations and show adaptation to local climatic conditions [15]. While previous studies assessed the factors related to amphibian population dynamics in the region [16,17,18,19,20], no quantitative estimates of long-term population trends are available for the *R. latastei* population of Monza Park. Given the recent lack of records of Italian agile frogs in the park, this study aimed to assess the long-term trend of this species in the park and how climate, alien species, and water management contributed to the observed temporal changes [16,17,18,20]. By combining data from more than 20 years of surveys, we quantified the dramatic decline and the possible local extinction of the species despite its legal protection [21].

## 2. Methods

### 2.1. Study Area and Study Species

The Monza Park is located in Northern Italy, in the Lombardy region (45.35 N, 9.16 E; Figure 1), and is a 750-ha urban park within the Lambro Valley Regional Park. The area (altitude: 162–200 m a.s.l.) is covered by fragmented forests and meadows. This park is known to host significant populations of the Italian agile frog, *Rana latastei*. As its boundary is approx. 10 km N of the municipality of Milan, it should represent the area closest to the type locality of the species [14,22].

The Italian agile frog is a small frog species that exhibits explosive breeding behavior. Mating season generally lasts a few days, typically spanning from mid-February to mid-April, depending on rainfall patterns and temperature regime [22]. Each female produces a single egg clutch per year (approx. 1200 eggs per clutch) [22,23]. Clutches are globular and are deposited onto submerged woods and tree roots within 1 m of the water surface [22]. Therefore, clutches can be easily spotted and recognized and their number corresponds to the number of breeding females in a population in a given season. Detection probability analyses showed that the detection probability of clutches is >95%; therefore, clutch counts enable reliable estimates of population trends [24]. Preferred sites for egg-laying are perennial and slow-flowing waterbodies, where hatching typically occurs within 2 or 3 weeks and is followed by a relatively long tadpole stage which, depending on water conditions, lasts about 2–3 months [22]. After metamorphosis, the froglets disperse into the landscape surrounding breeding sites. Sexual maturity is usually reached at 1 year, and young adults can colonize new wetlands. Adults generally show high site fidelity, continuing to breed in the same pond after the first season [25,26,27,28]. The maximum lifespan is 4 years, while the average lifespan is 1.7 years [28]. Despite the small range, several populations have developed local adaptations to local environmental conditions (e.g., climate and presence of predators); thus, some isolated populations have been proposed as evolutionarily significant units, deserving special conservation attention [15,23,29].

### 2.2. Field Monitoring and Data Analyses

Starting in 2000, we monitored seven freshwater sites (ponds and ditches with slow streams) in Monza Park (Figure 1) to assess the abundance of breeding *R. latastei* females by counting the number of egg clutches laid during the reproductive season. Field surveys were conducted in 10 different years, from 2000 to 2023: 2000, 2001, 2004, 2005, 2006, 2011, 2017, 2019, 2022, and 2023. Surveys were conducted at daytime, during the peak of the breeding season of the Italian agile frog (February–April) in order to perform visual counts of the egg clutches present at each site, following Dalpasso, Ficetola, Giachello, Lo Parrino, Manenti, Muraro, and Falaschi [20]. For each year, each site was surveyed multiple times (range: 3–7) between February and April. During surveys, we carefully inspected the whole surface of the sites and counted the number of egg clutches seen. The number of observers was usually two (range: 1–4). During a survey, each observer walked a section of the wetland, hence each portion of the wetland was explored only once during each survey. This allowed us to optimize time when more observers are available, without influencing detections. As we performed multiple clutch counts in the same year, we reported the highest count for each breeding site, which represents the minimum number of breeding females in each wetland [24]. In rare cases, we detected a few freshly laid egg clutches after the peak of the breeding season. These clutches were added to the highest count of that site, in order for the counts to better match the minimum number of females breeding in a given year. The minimum yearly abundance was then calculated as the sum of the highest count across all sites within a given year. We also recorded the presence/absence of the red swamp crayfish (*Procambarus clarkii*) and the presence or lack of water in each site across the entire study period. To prevent the spread of diseases, before and after each survey, all the material used, including shoes, was disinfected with 3% bleach [30].

### 2.3. Estimating Decline in Species Abundance

Several approaches can be used to assess species decline. According to the IUCN guidelines, a species decline can be evidenced by a decline in individual numbers and/or a decline in the area of occupancy and the extent of occurrence [31].

First, we used the number of clutches as a measure of the abundance of breeding females [24,32]. Amphibian populations can undergo large fluctuations in abundance over time, due to both deterministic and stochastic reasons [6,33]. Hence, population abundance in a single year may bear little information about long-term trends. Therefore, to obtain a more general perspective on the abundance of Italian agile frogs over time, we aggregated the counts on a base of 4-year periods. The average longevity of females for this species is about 3 years [28], thus using 4-year periods allows to limit potential fluctuations deriving from some females skipping reproduction in certain years (e.g., particularly dry periods). Samples were aggregated as follows: 2000/2003, 2004/2007, 2016/2019, 2020/2023; surveys were also performed during 2011, but during this year, we did not follow the standard protocol for counting clutches. Therefore, for 2011, we only reported the occurrence of clutches, without providing estimates of the overall abundance of the species. During 2020–2021, performing complete surveys was not possible due to travel restrictions related to COVID-19. We calculated the maximum clutch count per year and then the minimum and maximum abundance for each 4-year period, that is, the highest and lowest yearly counts, respectively. Counts were aggregated over 4-year periods only for graphical representation, while statistical analyses were performed with non-aggregated (year by year) data in which year was a continuous variable. 

Since the species was never found in the last period (see results), we ran a generalized linear mixed model (GLMM) to assess whether the population was showing any significant decline in abundance before its putative extinction in 2022. GLMM was run following Barker et al. [34]. Simulations and real-world analyses showed that GLMMs using observed abundances can provide good estimates of population trends even for species with imperfect detection [34]. In the GLMM, we used the maximum yearly clutch count in each site as a dependent variable, year as an independent variable, and site identity as a random effect. Before running the model, the year variable was scaled to mean = 0 and SD = 1 to improve model convergence. The number of clutches per site showed strong overdispersion; thus, GLMM was run in a Bayesian framework using the *brms* R package (version 2.19.0) with a negative binomial family [35]. The model was run with three chains for 4000 iterations, with 1000 iterations discarded as a burn-in and a thinning interval of 3, in order to sample 1000 posteriors for each chain. Convergence was ensured by Rhat values being <1.001.

Second, we estimated the range of the Italian agile frog within the park as the minimum convex polygon comprising all the breeding sites used by Italian agile frogs within a given 4-year period (extent of occurrence) [31,36]. Again, we considered a wetland occupied if breeding occurred during at least one year within each period. We then estimated declines as the proportional decline in the extent of occurrence.

### 2.4. Assessing the Factors Related to Changes in Occupancy

Given the small amount of data available, we used occupancy instead of abundance data to assess the factors related to the decline of the study species. We used a binomial GLMM to assess the relationship between the presence/absence of clutches and four variables representing climate and habitat suitability. For climate, we used mean temperature and total precipitation of March, which is the month corresponding to the peak of the breeding season of the species in the study area [37]. Climatic data were retrieved from the ERA5 database [38]. We downloaded the hourly temperature and precipitation for all the days of March of sampled years and, for each year, calculated the average temperature and the total precipitation of the month. Additionally, we considered two factorial variables related to human-mediated pressures on breeding sites: the occurrence of the invasive red swamp crayfish (0/1) and water management. The presence/absence of water in the study sites is managed by human activities, either through the regulation of water flow with artificial dams (sites D to G) or through the transformation of permanent waterbodies to dunghills (sites A to C). Therefore, we assigned a value of 0 when water was not present throughout the entire breeding season (inappropriate water management) and a value of 1 when water was present (appropriate water management).

We used the Integrated Nested Laplace Approximations (INLA) approach to fit the GLMM and estimate the posterior distribution of parameters. INLA represents an effective and powerful computational technique that allows running Bayesian models with complex spatial and temporal dependencies in alternative to Markov Chain Monte Carlo [39,40,41] and is very effective in detecting drivers of biodiversity change over time [42]. Continuous independent variables (temperature and precipitation) were scaled (mean = 0 and variance = 1) before analyses, and sub-population identity was included as a random factor. We used first-order autoregressive models to take into account temporal autocorrelation. Before running the analysis, we tested the correlation among independent variables and found no strong correlations (all Pearson’s correlation coefficients <|0.44|). The analysis was performed in the R environment using the INLA package [40] (version 23.04.24).

## 3. Results

At the beginning of our study, Italian agile frogs laid clutches in up to seven wetlands spread across the whole park (Figure 2).

### 3.1. Number of Breeding Females

The number of clutches per site showed high variability among sites and years, ranging from 0 to 207 (Figure 2). The abundance of the frog was highly skewed across sites, with just one site (site D in Figure 1) accounting for >50% of clutches in all years except 2004. During some years, >95% of breeding females laid clutches in that specific site (Figure 2).

Besides the last two years, the total number of clutches across all sites ranged from 45 to 224 (Figure 2). Within each 4-year period, the number of breeding females varied between a minimum of 45–72 to a maximum of 192–224 (Figure 3). Despite variation between adjacent years and some local extinction/recolonization, the overall population size appeared to be rather stable until 2016–2019. The generalized linear model used to assess the possible decline of the species before its putative extinction from the study area showed an average regression coefficient between year and abundance of −0.04 (95% credible interval (CI): −0.16 to 0.08), suggesting that no evidence of population decline was detectable prior to the disappearance of the species from the study area (Table 1).

Despite repeated surveys, no frogs or frog clutches were detected in 2022 and 2023, suggesting that the species went extinct in the park after 2019.

### 3.2. Extent of Occurrence within the Park

At the beginning of the study, frogs occurred both in Northern and Southern sites (Figure 1 and Figure 4), with a total extent of occurrence of 0.73 km^2^. However, between 2000 and 2001, the three northernmost sites (A, B, C in Figure 1) were destroyed with the conversion of the small waterbodies into dunghills, leading to the complete disappearance of frogs in the area. The destruction of these sites determined an 88% decline in the extent of occurrence. The extent of occurrence remained stable from 2004 to 2019 but was then followed by the extinction of all the populations in the southern portion of the park (Figure 4).

### 3.3. Environmental Factors Driving Occupancy Changes

The model assessing the effect of climatic and habitat variables on changes in site occupancy showed no evidence of an effect of climatic features, with the posterior distribution for both temperature and precipitation parameters overlapping zero (Figure 5). Water management was the factor most strongly related to the presence of clutches sites, with a very strong positive effect. This indicates that, if water management allowed water presence during the breeding season, sites were usually occupied (Figure 5). The presence of the invasive crayfish showed a negative effect, meaning that Italian agile frogs are more likely to lay eggs in sites without the presence of the crayfish. While the 95% CI for crayfish presence did not overlap zero, the strength of its effect was much smaller compared to the effect size of water management (Figure 5).

## 4. Discussion

Our long-term monitoring study assessed the temporal changes in the abundance of the Italian agile frog, detecting its possible local extinction from an area that was probably the closest known occurrence to the type locality.

When we used the number of clutches as an estimate of abundance, the population of Monza Park showed a stable trend from the beginning of our long-term sampling until 2019 (Figure 2 and Figure 3), with the number of clutches consistently fluctuating between 40 and >200 (Figure 3). Then, in 2022, the species completely disappeared from all the study sites, and also, in 2023, neither clutches nor adults were found during field surveys (Figure 3). The number of clutches was clearly not stable between 2000 and 2019, as it showed striking oscillation between consecutive years. Nevertheless, years with few active females (and thus, presumably, poor breeding success) were generally followed by years with a much larger number of breeding females, and some local extinctions were followed by recolonizations, as expected under meta-population dynamics [19,43]. For instance, in 2004, just 49 clutches were detected, but this value showed a four-fold increase in 2005. Strong variations in the number of breeding amphibians are well known to occur and are generally related to the interplay between habitat variability, weather conditions, and stochastic demographic factors [6,43,44,45]. Weather conditions and water availability are key determinants of variation in the number of breeding individuals [45]; as during dry years, lower water availability in wetlands not only limits the number of breeding adults but also affects larval stage mortality by increasing desiccation risk [20,46]. The effects of these oscillations can be exacerbated in small and isolated populations. Unfortunately, Monza Park is severely isolated from other sites hosting Italian agile frogs. The nearest known populations occur ~15 km from the park. Given that the known dispersal distance of these frogs is not expected to exceed 1 km [47], dispersal events between this park and nearby populations are highly unlikely, as confirmed by population genetic analyses [48].

Nevertheless, the number of active individuals is just one possible parameter providing estimates of population size. Measuring population reduction on the basis of the extent of occurrence provided much earlier evidence of decline (Figure 4). This occurred because habitat loss due to the expansion of farming activities determined the loss of three breeding sites, i.e., nearly 50% of all the breeding sites. Even if these sites were small and did not account for a large number of clutches (Figure 2), their unsuitability strongly reduced the range of the species within the park. Given the limited dispersal ability of amphibians, this can have long-term effects with crucial impacts on population dynamics. For instance, small isolated populations can rescue spatially structured populations if the largest source populations become temporarily unsuitable (see below for a specific discussion).

The causes of the possible local extinction of the Italian agile frog from Monza Park are complex, and this extinction was somewhat unexpected given the stable trend of the number of breeding females between 2000 and 2019. The local extinction of this species has also been reported in other core areas of its range. For instance, a recent study [20] documented the disappearance of this species from the Groane Regional Park, probably due to high sensitivity to drought, short lifespan, and demographic stochasticity. Our model (Figure 5) showed that the occurrence of egg clutches was strongly related to the persistence of water and the absence of invasive crayfish. Therefore, the possible extinction of the species from the park may be related to the joint effect of inappropriate management (increase in human-mediated droughts) and the invasion by the red swamp crayfish. Many amphibian populations, including the Italian agile frog, survive within meta-population networks [19,43]. Our data do not have the temporal completeness needed to develop explicit models of spatially structured populations [19]. The concentration of most breeding individuals in a single pond (Figure 2) is in agreement with the idea that this site generally acts as a source within a meta-population [49]. Unfortunately, the unsuitability of source sites for a few years can have dramatic impacts on the dynamics of the overall metapopulations, especially in short life-span species, such as *R. latastei*. Additionally, the invasive red swamp crayfish (*Procambarus clarkii*) was introduced in the area around 2005, possibly causing further impacts on this frog, as this crayfish exerts strong predation pressure on amphibian tadpoles [16,18].

The Monza Park is actively managed, and during spring 2021, the main breeding sites were without water because of maintenance work. Subsequently, the whole study area was subject to a severe drought between winter 2022 and early spring 2023 [50]. The appropriate management of waterbodies is a crucial task to ensure the persistence of amphibian populations in human-dominated landscapes. The persistence of water in breeding sites during spring is essential to ensure the complete development of amphibian larvae, and several methods are available to provision water, among which the most effective are the redirection of water through flow-regulating structures and water pumping [51,52]. However, when a certain region is subject to drought, water use for the conservation of biodiversity and for human use (mainly agriculture) may collide [53]. Therefore, in areas where drought events are expected to be more frequent in the future, the optimization of water use for agriculture and other purposes should also consider the importance of maintaining water resources to ensure the correct development of aquatic and semi-aquatic organisms. This is essential should we want to avoid losses of irreplaceable pieces of biodiversity [54].

Our analyses showed that even in the absence of a decline, populations that were self-sustaining over a long period can rapidly decrease and even disappear (Figure 3). The local extinction of a population can be particularly problematic in suburban areas, where the natural environment is more fragmented, and barriers created by urban and agricultural lands can prevent the recolonization of unoccupied sites [55]. This means that, after dealing with the possible factors that caused the local extinction of a population, human-mediated translocations would be required to re-establish a self-sustaining population in an isolated area, such as the Monza Park. However, previous analyses have shown that the long-term isolation of the Monza Park population determined the development of striking local adaptations that allow them to withstand the local environmental challenges. For instance, these populations evolved a fast development rate in response to cold microclimate and the occurrence of alien predators [15,23,29]. As the persistence of the Italian agile frog is strongly related to these adaptations, the re-introductions of individuals belonging to the same species, but lacking specific adaptations, can be challenging. We, therefore, stress that biodiversity-friendly practices (e.g., appropriate water management and preservation of ecologically relevant sites) should be paired with regular monitoring of biodiversity, in order to promptly identify threats and implement appropriate and timely conservation interventions, and prevent local extinctions.

In principle, the results of our sampling do not necessarily imply a complete extinction of the Italian agile frog from Monza Park, since some adult individuals might be still present in the available terrestrial habitats, or we might have missed a very small fraction of clutches. Nevertheless, this is rather unlikely, given the high monitoring efforts to which the park is subjected and considering the high detectability and distinctiveness of *R. latastei*. This putative extinction is probably related to the joint effect of inappropriate management and severe droughts. While future surveys are still needed to verify the actual local extinction of the Italian agile frog, we point out that eventual reintroduction efforts should be closely followed by appropriate management of the territory by local authorities, to avoid repeating the observed drastic decline that would erase undermine conservation action.

## 5. Conclusions

*Rana latastei* is currently listed as vulnerable by the national, European, and global IUCN redlists because it has a small area of occupancy and suffers severe fragmentation and continuing population decline [13,56,57]. The observed decline and extinction within Monza Park confirm the continuing decline in the number of locations reported by the IUCN for more than 30 years [13] and reiterates the need for conservation measures for this endemic frog. The EU Habitats Directive (Directive 92/43/EEC) is one of the major tools for the conservation of biodiversity in Europe, as it lists more than 1200 strictly protected species for which habitats and breeding sites also need to be protected. The Habitats Directive aims to “ensure the long-term survival of Europe’s most valuable and threatened species and habitats” [58] and has developed several tools to halt the decline of listed species. Despite 30 years of efforts, a large proportion of listed species are experiencing a worsening of their status [21]. We join recent calls for the development of more effective and stringent tools for the conservation of European biodiversity.

## Figures and Tables

**Figure 1 animals-13-03187-f001:**
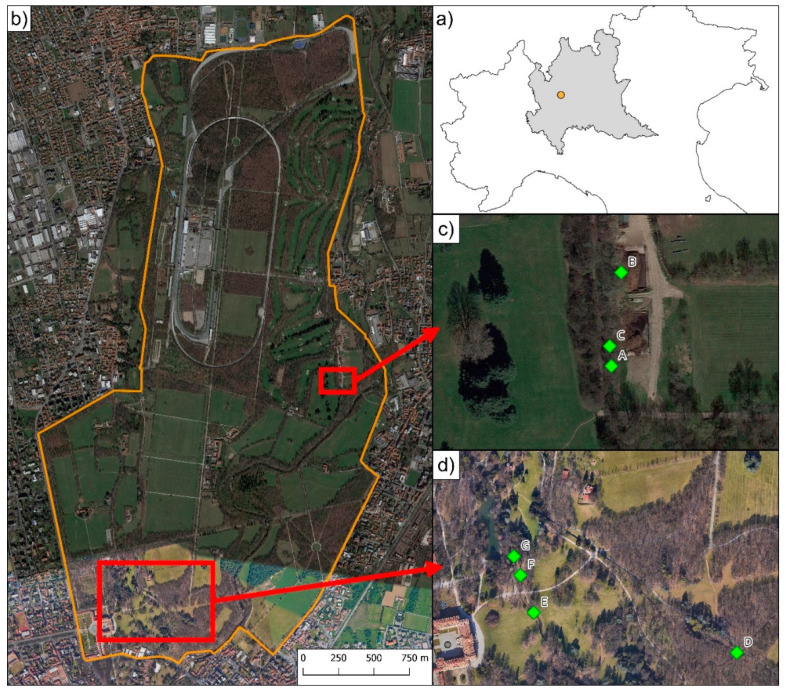
Location of the study sites in Monza Park (Northern Italy). (**a**) Location of Monza Park in the Lombardy region (grey) within Northern Italy. (**b**) The two main areas (red squares) hosting breeding sites for *Rana latastei* in Monza Park are detailed in panels (**c**,**d**). Letters A–G indicate the site codes.

**Figure 2 animals-13-03187-f002:**
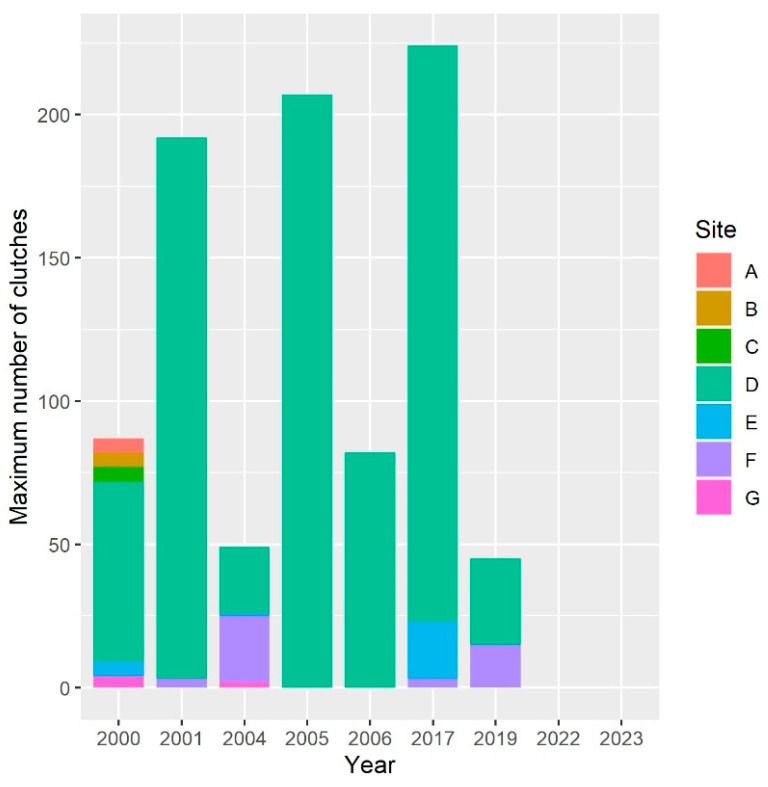
Counts of Italian agile frog’s egg clutches present at breeding sites in Monza Park between 2000 and 2023. For each site, the maximum number of clutches registered for each year is reported. Site codes are provided in Figure 1.

**Figure 3 animals-13-03187-f003:**
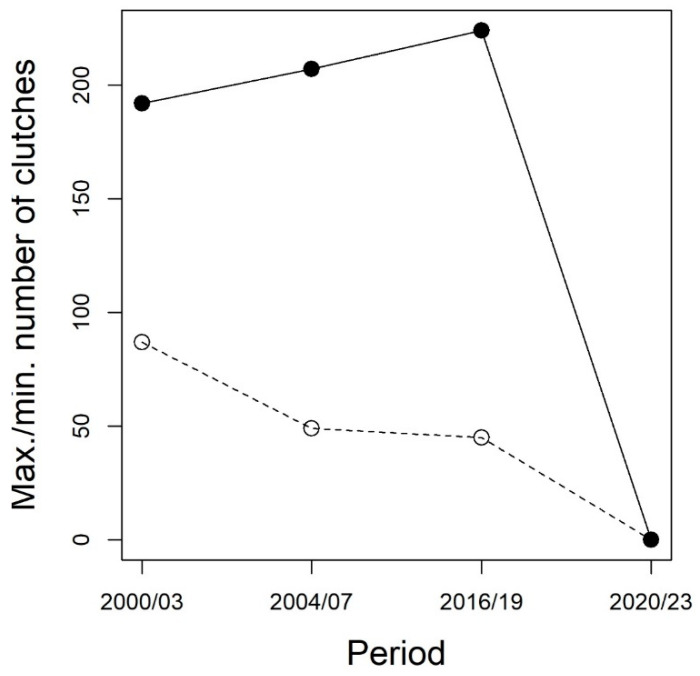
Maximum (black dots) and minimum (empty dots) number of Italian agile frog clutches in four 4-year periods in Monza Park. Within each period, the minimum and maximum values were calculated as the highest and lowest yearly abundances; therefore, they are an indication of the range of abundance within each 4-year period.

**Figure 4 animals-13-03187-f004:**
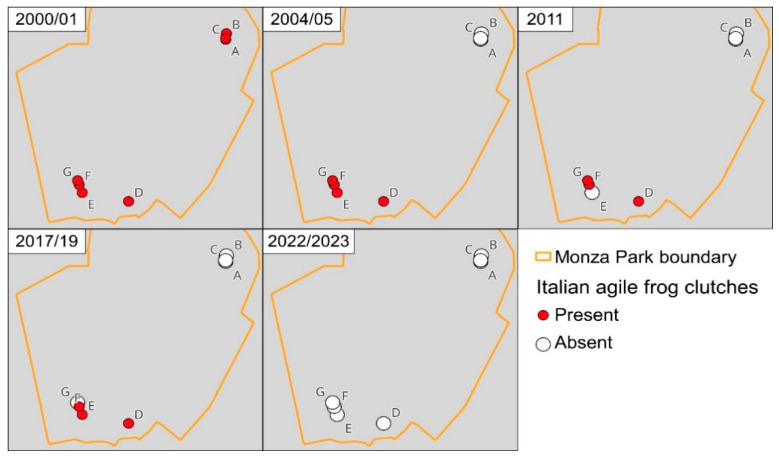
Changes in the distribution of the Italian agile frog within Monza Park from 2000 to 2022. For each period, red dots indicate sites where frog clutches were observed, and white dots indicate sites where clutches were not observed. Letters A–G indicate the site codes.

**Figure 5 animals-13-03187-f005:**
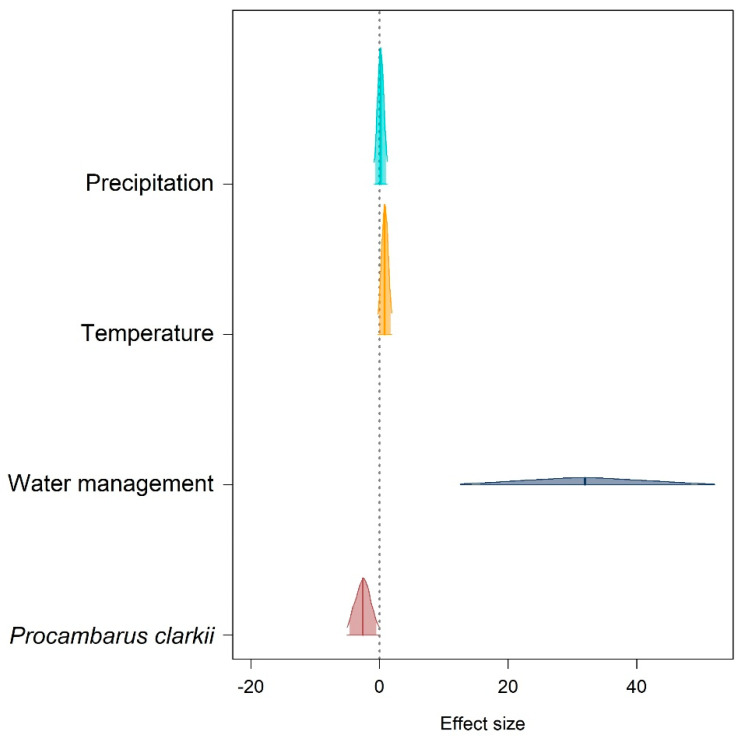
Posterior distribution density of the parameters estimated by the model assessing the environmental factors related to changes in occupancy. For each parameter, the vertical line is the median, the outer line represents the 95% credible interval, and the shaded area represents the 90% credible interval.

**Table 1 animals-13-03187-t001:** Parameters estimated by the model assessing the trend in abundance of *Rana latastei* in Monza Park between 2000 and 2019. For each parameter, the mean, standard error, and 95% credible interval of the posterior distribution are shown.

Parameter	Mean Estimate	Standard Error	Lower 95% CI	Upper 95% CI
Intercept	0.73	1.13	−1.76	2.78
Centered year	−0.04	0.06	−0.16	0.08
Site random effect SD	2.23	0.88	1.02	4.45

SD = standard deviation; CI = credible interval.

## Data Availability

All data and scripts are available on Figshare (https://figshare.com/s/5b6f23ffca0818df547b (accessed on 8 October 2023)).

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
