# Peer review of "Decline and Extinction of the Italian Agile Frog Rana latastei from Core Areas of Its Range"

_animals, 2023, doi:10.3390/ani13203187_

Round 1

Reviewer 1 Report

I am not fully convinced about extinction. It needs one or two years monitoring more as climatic conditions may have an impact.

It's a very small (geographically) study restricted to one park.  It should be useful to get more informations on other locations of the species and to make a "review" of population status on the range distribution

the discussion is long and explore hypothesis without testing them. It should be useful to analyse, as an example meteorological data to test at least the influence of weather

Author Response

I am not fully convinced about extinction. It needs one or two years monitoring more as climatic conditions may have an impact.

RESPONSE: We agree that monitoring should continue at least for two more years, indeed, plan to perform more surveys in 2024-2025. In the discussion we pointed out that in principle the species might be still present, given that we may have missed some adults or a very small portion of egg clutches (see lines 371-381). However, we stress the importance of prompt publication of these issues, as these reports are fundamental to speed up conservation actions

It's a very small (geographically) study restricted to one park.  It should be useful to get more informations on other locations of the species and to make a "review" of population status on the range distribution

RESPONSE: We thank the reviewer for the suggestion. Unfortunately, long term data covering the whole range of the species do not exist. In fact, also the IUCN redlists only has access to long-term data for a few localized populations:

https://www.iucnredlist.org/species/19156/215527729#assessment-information.

We believe this lack of data makes our manuscript particularly relevant

the discussion is long and explore hypothesis without testing them. It should be useful to analyse, as an example meteorological data to test at least the influence of weather

RESPONSE: We thank the reviewer for constructive comment. In this revision we present an additional analysis, assessing the effect of both climatic and habitat variables on the presence/absence of egg clutches in study sites. We show that water management and the presence of invasive crayfish are the main drivers of occupancy (see Fig. 5). Following the reviewer’s suggestion, we modified the discussion to implement a more robust testing of the hypotheses (lines 202-208, 267-282, 324-329).

Reviewer 2 Report

Decline and extinction of the Italian agile frog Rana latastei from core areas of its range

Overall

The paper aims to provide insight into a possible local extinction event of the Italian agile frog. The difficulty of reviewing papers sometimes is to ensure you review what the paper versus what the paper could be. I have tried to do this however I want to point out to the authors that they may wish to retract their submission and add environmental variables to the mix. We are at a stage in which just documenting declines is not useful as we need to know why they are happening. By not incorporating any environmental parameters in this study the study can only be considered a research note i.e. it documents the unfortunate local extinction. The paper cannot be sold as it is as a monitoring of a decline with the soft sell of offering some reasonings for the decline. In its current state I can only see one option of increasing the impact of the work which is highlighting that the use of count data alone is not useful for species that rely on certain favourable environmental conditions to breed. But as the authors themselves indicated without environmental data shifts in abundance can be hard to distinguish between natural variation and actual declines.

Before turning to the specific aspects of the paper itself, I want to acknowledge that although there seems to be a decent dataset from a period of 20 years (although note with gaps of years with no sampling it seems), I cannot understand why there is a lack of parameters that could have been used to accompany the count data. Surely you can incorporate some environmental parameters? Were there not protocols for measuring temperature, water quality etc and if not some of these can be accrued retrospectively. Declines in numbers as indicated in the introduction requires taking into account fluctuations of normal seasonal responses. So, if the frogs are known to respond to certain natural environmental conditions, then that needs to be incorporated into the models so you remove the variation related to seasonal differences alone. Once that is controlled for, the remaining residuals should give you some insight into whether a notable decline was detectable or not prior to them disappearing. If the purpose of the study though was to ascertain if counts is enough to alert monitors of an imminent extinction risk, then you need to be upfront and clear about this in the aims and conceptual framework. At the end of the day incorporating even some temperature variables would have been better than nothing. Given the current conservation crisis just acknowledging something is going extinct is not particularly valuable anymore but trying to understand what is driving it or may have caused it. So going that extra mile would make this paper more worthwhile. So just getting some climate data retrospectively to incorporate with your modelling would be great. As it stands the paper is a document of a disappearance with little else to offer except that count data is not enough it seems. So you have to either pose this paper as a note, that is, a document of a potential extinction event with count and trends and try not to make any more of the results than that or you go away and add environmental data for increased impact.

Introduction

(Overall the introduction could do with some minor editing by a native English speaker there are a number of tense errors (as present vs past tense)

Line 43 – replace ‘has been’ with ‘was’

I suggest rewriting the first paragraph – firstly its not actually a paragraph as it has no middle component. I suggest placing a full stop after 1-3. Then stating amphibians of all vertebrates (remember invertebrate are animals too and some of their declines have been far greater than amphibians) have shown rapid decline and provide some evidence in support of this within the paragraph.

Line 45 – who is ‘we’?

Line 50 Rican not Rica

Line 51 – 54 needs to be two sentences it has two incomplete thoughts – in fact the whole paragraph could do with some editing and better story structure as it was not totally clear what point was being made. I think it was trying to indicate that in the case of the golden frog a lack of monitoring of the species meant it went extinct without having any clear understanding of what may have lead to the decline?? I also do not understand what the monitoring efforts had to do with aiding the designation of critically endangered. The text gives the impression that the monitoring should have reversed the endangerment. But monitoring is just counting it is not a conservation intervention in itself.  

Line 67-69: needs to be rewritten – when you say surveyed you are referring to someone just going to see if the amphibians are there on occasion? Rather than an ongoing monitoring regime i.e. taking details on status at the same time each year? If that is the case then you should be saying General adhoc surveys of amphibians have been conducted in Northern Italy since the 1990’s. Regular monitoring has only occurred since 2009 in which data pertaining to their status such as population size etc etc etc ### has been collected.

Line 69-71: too vague this line could be said about any specie sin the entire world

Line 71-72: but did you just say there had been monitoring since 2009? You need to be cleared on your locality boundaries here and why does it matter if the monitoring has been done closer to milan city? The way it is written it just seems this area was chosen as it is convenient rather than having a solid a priori reasoning for its selection. (ok just read the aims and I now understand – I would couple the type locality with the above reference to Milan city rather than just in the aims. Also, because it is the type locality it does not necessarily make it the best location for the species.

Lines 73-77: Given the authors have not provided any indication of whether other variables were collected during the surveys such as environmental and/or conditions such as water quality or veg cover this aim needs to be modified. The aim, to make this study valid as is, needs to be about the use of abundance data alone to detect potential extinction risk. So you either change the aim or clarify the aim with a conceptual framework ie hypotheses and predictions and/or indicate the limited scope of the study from the outset. That is announce the study is examining if extinction risk in this species could have been detected through abundance data and/or occurrence alone.It is already well known that if the species are a slave to certain environmental conditions before breeding will occur then using abundance only is unlikely to be effective hence you typically need to model some environmental parameters such as temp or water availability. Your model has to control for the natural fluctuations before you can detect any downward or upward trend.

Methods

Ok what is not quite clear is this relationship between highest clutch counts, minimum number of breeding pairs and the multiple monitoring – I am assuming that if the egg clutches are found in a different location though out the multiple monitoring period that it is added as a new female? So is the total the number of egg clutches found within unique micro-localities across the entire season of monitoring? You suggest the highest count – this suggests you just selected the highest count among the multiple counting events. If that is true then you may need to note a caveat that this may not represent all of the breeding females especially if there is a delay in some individuals breeding relative to others?? I am assuming this is what you mean by representing the minimum amount of breeding females? I think just being more explicit here would be good.

This clustering of counts across a four-year cycle seems a bit odd to me. I kind of get why you have done it as it’s a rolling average and thus smooths out the trend line. But you lose information and the variable becomes a category and cannot be treated as a count as such. Like in the first four years they used multiple ponds but was that in each of those years or did only one of those years show this trend and the others did not? Actually, looking at this again you have uneven sampling within these arbitrary ranges so some have sampling over 2 years some over 3 years and the 4-year cycle idea only makes sense if in 2000 all the frogs were the same age. So, I am not certain this is the right approach. But you have been transparent in what you have done which is good.

Line 113-114: ‘we carefully inspected the whole surface of sites’ how did you standardise this? This description is not useful for someone trying to replicate the study or the protocol. Was the survey done with multiple people did you cover a certain area over a certain time or at a certain rate etc etc – I suspect not as we generally do not take such measures when we should but if you did please incorporate as the descriptor is actually not that helpful. So you can say we used an adhoc design (unless you have a transect or stratified approach of course) in which we walked along the end of the ponds (?) at approximately # rate stopping only to count clutches. And the approach taken how did you standardise it from each sampling period?

Line 121: Remove ‘For instance’ and Replace with ‘The IUCN accepts a suggestion of a species decline if evidenced by a decline in individual numbers, and/or a decline in the area of occupancy and the extent of occurrence’

Line 146: year variable scaled by the mean? I do not understand how this is possible given the year variable is categorial isn’t it as you have converted the years to a cluster of years?

The map in the methods seems to have results statements like the northern region disappearing etc I think for the methods you just need to show the sites and the overall locality.

Results

If you are running a Bayesian model you need to also present some other figures and parameters I believe. I am not an expert on Bayesian modelling but I do recognise that there needs to be further displays of the modelling outputs either presented in the text or as supplementary material.

Discussion

Line 308: ‘join teffect’ should be ‘joint effect’

A better understanding of the site fidelity of these frogs would be good like is expected they move between these ponds – given the variation in sites and fluctuation of use is it possible the females are moving around more than might be expected?

Ponds G and E something happened in 2011 and 2017 periods – they were not using E in 2011 but were G and in 2017 that flipped why?

Overall the english is fine but there was notable areas of the need for a further look over. The introduction needs an english edit quite a number of errors with regards to past and present tense. Also quite a number of the statements are too vague in the introduction. The methods and results were better written while the discussion could deal with a minor edit.

Author Response

The paper aims to provide insight into a possible local extinction event of the Italian agile frog. The difficulty of reviewing papers sometimes is to ensure you review what the paper versus what the paper could be. I have tried to do this however I want to point out to the authors that they may wish to retract their submission and add environmental variables to the mix. We are at a stage in which just documenting declines is not useful as we need to know why they are happening. By not incorporating any environmental parameters in this study the study can only be considered a research note i.e. it documents the unfortunate local extinction. The paper cannot be sold as it is as a monitoring of a decline with the soft sell of offering some reasonings for the decline. In its current state I can only see one option of increasing the impact of the work which is highlighting that the use of count data alone is not useful for species that rely on certain favourable environmental conditions to breed. But as the authors themselves indicated without environmental data shifts in abundance can be hard to distinguish between natural variation and actual declines.

Before turning to the specific aspects of the paper itself, I want to acknowledge that although there seems to be a decent dataset from a period of 20 years (although note with gaps of years with no sampling it seems), I cannot understand why there is a lack of parameters that could have been used to accompany the count data. Surely you can incorporate some environmental parameters? Were there not protocols for measuring temperature, water quality etc and if not some of these can be accrued retrospectively. Declines in numbers as indicated in the introduction requires taking into account fluctuations of normal seasonal responses. So, if the frogs are known to respond to certain natural environmental conditions, then that needs to be incorporated into the models so you remove the variation related to seasonal differences alone. Once that is controlled for, the remaining residuals should give you some insight into whether a notable decline was detectable or not prior to them disappearing. If the purpose of the study though was to ascertain if counts is enough to alert monitors of an imminent extinction risk, then you need to be upfront and clear about this in the aims and conceptual framework. At the end of the day incorporating even some temperature variables would have been better than nothing. Given the current conservation crisis just acknowledging something is going extinct is not particularly valuable anymore but trying to understand what is driving it or may have caused it. So going that extra mile would make this paper more worthwhile. So just getting some climate data retrospectively to incorporate with your modelling would be great. As it stands the paper is a document of a disappearance with little else to offer except that count data is not enough it seems. So you have to either pose this paper as a note, that is, a document of a potential extinction event with count and trends and try not to make any more of the results than that or you go away and add environmental data for increased impact.

RESPONSE: We thank the reviewer for constructive comment. Following the reviewer’s comment, in this revision we present an additional analysis, assessing the effect of both climatic and habitat variables on the presence/absence of egg clutches in study sites. Specifically, we added 1) two variables describing climatic features of the area (mean temperature and total precipitation of March in different years), one variable representing the invasion by alien invasive species (introduction of the red swamp crayfish) and one variable representing the management of water during the specie’s breeding season.

Through the use of Bayesian models, we show that water management and the presence of invasive crayfish are the main drivers of occupancy (see Fig. 5). We expanded the methods, results, and discussion accordingly (lines 134-136, 180-208, 237-277, 324-329).

Introduction

(Overall the introduction could do with some minor editing by a native English speaker there are a number of tense errors (as present vs past tense)

REPONSE: We carefully revised the introduction correcting tense errors.

Line 43 – replace ‘has been’ with ‘was’

RESPONSE: Done.

I suggest rewriting the first paragraph – firstly its not actually a paragraph as it has no middle component. I suggest placing a full stop after 1-3. Then stating amphibians of all vertebrates (remember invertebrate are animals too and some of their declines have been far greater than amphibians) have shown rapid decline and provide some evidence in support of this within the paragraph.

RESPONSE: We modified the paragraph as suggested (lines 43-49).

Line 45 – who is ‘we’?

RESPONSE: We changed “we” to “conservationists”.

Line 50 Rican not Rica

RESPONSE: We changed as suggested.

Line 51 – 54 needs to be two sentences it has two incomplete thoughts – in fact the whole paragraph could do with some editing and better story structure as it was not totally clear what point was being made. I think it was trying to indicate that in the case of the golden frog a lack of monitoring of the species meant it went extinct without having any clear understanding of what may have lead to the decline?? I also do not understand what the monitoring efforts had to do with aiding the designation of critically endangered. The text gives the impression that the monitoring should have reversed the endangerment. But monitoring is just counting it is not a conservation intervention in itself.

RESPONSE: Following the reviewer’s comment, we rephrased the paragraph to better point out that, without monitoring, a species can go extinct without having any clear understanding of what may have lead to the decline, and without any attempt of halting / reversing the decline (lines 61-63).

Line 67-69: needs to be rewritten – when you say surveyed you are referring to someone just going to see if the amphibians are there on occasion? Rather than an ongoing monitoring regime i.e. taking details on status at the same time each year? If that is the case then you should be saying General adhoc surveys of amphibians have been conducted in Northern Italy since the 1990’s. Regular monitoring has only occurred since 2009 in which data pertaining to their status such as population size etc etc etc ### has been collected.

Line 69-71: too vague this line could be said about any specie sin the entire world

RESPONSE: These sentences were deleted as a result of the changes made to the paragraph following the comments below.

Line 71-72: but did you just say there had been monitoring since 2009? You need to be cleared on your locality boundaries here and why does it matter if the monitoring has been done closer to milan city? The way it is written it just seems this area was chosen as it is convenient rather than having a solid a priori reasoning for its selection. (ok just read the aims and I now understand – I would couple the type locality with the above reference to Milan city rather than just in the aims. Also, because it is the type locality it does not necessarily make it the best location for the species.

Lines 73-77: Given the authors have not provided any indication of whether other variables were collected during the surveys such as environmental and/or conditions such as water quality or veg cover this aim needs to be modified. The aim, to make this study valid as is, needs to be about the use of abundance data alone to detect potential extinction risk. So you either change the aim or clarify the aim with a conceptual framework ie hypotheses and predictions and/or indicate the limited scope of the study from the outset. That is announce the study is examining if extinction risk in this species could have been detected through abundance data and/or occurrence alone.It is already well known that if the species are a slave to certain environmental conditions before breeding will occur then using abundance only is unlikely to be effective hence you typically need to model some environmental parameters such as temp or water availability. Your model has to control for the natural fluctuations before you can detect any downward or upward trend.

RESPONSE: Following the comments above, we rephrased the entire paragraph to better point out the aim of the study and avoid unclear sentences (L 72-82). Our aim was to assess the long-term trend (and now also the factors related to these trend) of R. latastei in a protected area close to its type locality and holding unique genetic features (e.g. Melotto et al. 2020 Nature Communications).

Methods

Ok what is not quite clear is this relationship between highest clutch counts, minimum number of breeding pairs and the multiple monitoring – I am assuming that if the egg clutches are found in a different location though out the multiple monitoring period that it is added as a new female? So is the total the number of egg clutches found within unique micro-localities across the entire season of monitoring? You suggest the highest count – this suggests you just selected the highest count among the multiple counting events. If that is true then you may need to note a caveat that this may not represent all of the breeding females especially if there is a delay in some individuals breeding relative to others?? I am assuming this is what you mean by representing the minimum amount of breeding females? I think just being more explicit here would be good.

RESPONSE: The reviewer is correct; we show the highest count among multiple count events. However, our surveys were timed in order to match the peak of the breeding activity of the species. We agree that this may not represent all the breeding females if a delay of some individuals occurs, however, surveys were performed also after the peak of the breeding season. Rana latastei is an explosive breeder, with nearly all t he females laying eggs within one week (Seglie et al., 2008. Comportamento riproduttivo e vocale della rana di Lataste, Rana latastei (Amphibia: Anura). In: Corti, C. (Ed.), Herpetologia Sardiniae, Belvedere, Latina, pp. 439-443). As clutches require 2-3 weeks before hatch, a few clutches laid after the peak are not expected to affect our conclusions. This information is available at L 94-97. In very few cases we detected one or two fresh egg clutches later in the season, and these were added to the highest count in order for the counts to better match the minimum number of females breeding in a given year. Following the reviewer’s comment, we implemented this explanation in the methods (lines 130-133).

This clustering of counts across a four-year cycle seems a bit odd to me. I kind of get why you have done it as it’s a rolling average and thus smooths out the trend line. But you lose information and the variable becomes a category and cannot be treated as a count as such. Like in the first four years they used multiple ponds but was that in each of those years or did only one of those years show this trend and the others did not? Actually, looking at this again you have uneven sampling within these arbitrary ranges so some have sampling over 2 years some over 3 years and the 4-year cycle idea only makes sense if in 2000 all the frogs were the same age. So, I am not certain this is the right approach. But you have been transparent in what you have done which is good.

RESPONSE: Yes, we clustered the counts to show a smoothed trend in the minimum/maximum number of clutches showed in Fig. 3. The aggregation over four-year cycles was just for graphical representation, while for all the statistical analyses we used non-aggregated data to take into account fluctuations in abundance. Following the reviewer’s suggestion, we specified this in the methods to avoid misunderstandings (lines 158-160).

Line 113-114: ‘we carefully inspected the whole surface of sites’ how did you standardise this? This description is not useful for someone trying to replicate the study or the protocol. Was the survey done with multiple people did you cover a certain area over a certain time or at a certain rate etc etc – I suspect not as we generally do not take such measures when we should but if you did please incorporate as the descriptor is actually not that helpful. So you can say we used an adhoc design (unless you have a transect or stratified approach of course) in which we walked along the end of the ponds (?) at approximately # rate stopping only to count clutches. And the approach taken how did you standardise it from each sampling period?

RESPONSE: We added more details on how we standardized the sampling. In particular “The number of observers was usually two, varying from one to four. During a survey, each observer walked a section of the wetland, hence each portion of the wetland is explored only once during a single survey. This allows to optimize time when more observers are available, without influencing detections.” (lines 124-128)

Line 121: Remove ‘For instance’ and Replace with ‘The IUCN accepts a suggestion of a species decline if evidenced by a decline in individual numbers, and/or a decline in the area of occupancy and the extent of occurrence’

RESPONSE: We modified the sentence as suggested (lines 123-125)

Line 146: year variable scaled by the mean? I do not understand how this is possible given the year variable is categorial isn’t it as you have converted the years to a cluster of years?

RESPONSE: We think this misunderstanding is a consequence of the lack of clarity of the aggregation over four-year periods. Indeed, in this model we used non-aggregated data, year was a continuous variable, and centring independent variables is a common practice to increase convergence and avoid overly large estimation of intercept values. We think that after clarifying the reason of aggregation into four-year periods (lines 158-160), it is now clear that this analysis involved non-aggregated (year to year) data.

The map in the methods seems to have results statements like the northern region disappearing etc I think for the methods you just need to show the sites and the overall locality.

RESPONSE: We modified the caption to avoid result-like statements (lines 211-213).

Results

If you are running a Bayesian model you need to also present some other figures and parameters I believe. I am not an expert on Bayesian modelling but I do recognise that there needs to be further displays of the modelling outputs either presented in the text or as supplementary material.

RESPONSE: The reviewer is absolutely correct. Following this suggestion, we added a table showing the parameters estimated by the model (Table 1), which was missing in the first version of the manuscript. Thank you very much for noticing.

Discussion

Line 308: ‘join teffect’ should be ‘joint effect’

RESPONSE: We corrected the typo, thank you for noticing.

A better understanding of the site fidelity of these frogs would be good like is expected they move between these ponds – given the variation in sites and fluctuation of use is it possible the females are moving around more than might be expected?

RESPONSE: Usually adults show high site fidelity. In particular, after metamorphosis the froglets disperse into the landscape surrounding breeding sites. The sexual maturity is reached usually at the age of one year and young adults can colonize new wetlands. Adults generally show high site fidelity, continuing to breed in the same pond after the first season. The maximum lifespan is four years, while the average lifespan is 1.7 years. We added these details into the methods (lines 105-109).

Ponds G and E something happened in 2011 and 2017 periods – they were not using E in 2011 but were G and in 2017 that flipped why?

RESPONSE: As many amphibians, Rana latastei shows a typical meta-population pattern, with local local extinction and re-colonization in “sink” sites that are connected to “source sites”. This has been described in depth for larger metapopulation systems (Manenti et al., 2020. Ecography 43, 119-127). So the most likely explanation is that in 2017 E was re-colonized by juveniles originating from F. We slightly expanded the results (L 227) to highlight these extinctions / recolonizations. Furthermore, in the discussion we better highlight the importance of metapopulation dynamics for amphibian populations (L 295-296)

Comments on the Quality of English Language

Overall the english is fine but there was notable areas of the need for a further look over. The introduction needs an english edit quite a number of errors with regards to past and present tense. Also quite a number of the statements are too vague in the introduction. The methods and results were better written while the discussion could deal with a minor edit.

RESPONSE: Following the reviewer’s suggestion, we carefully revised the English posing particular attention to past and present tense.

Reviewer 3 Report

The authors present the results of a long-term monitoring on the vulnerable species Rana latastei in the Monza Park and document the lack of clutches in the last years, concluding that the species went extinct in the study area. The paper is well written and the data are clearly presented to support the conclusions. There are some points that should be better explained: 1) being the frog vulnerable, authors should have permissions from local authorities to study the species and cite them in the text; 2) for the same reasons, details about field procedures and mitigating measures should be presented, e.g., disinfection of the equipment such as boots or gloves 3) as an example of recent extinction of anurans authors cite in the introduction the Costarican golden toad Incilius periglenes. Among the possible causes of the extinction the pathogenic chytrid fungus Batrachochytrium dendrobatidis has been hypothesized. Did the authors consider the fungus as a possible cause of disappearence R. latastei? 4) The authors monitored the presence of clutches as an estimator of decline and regarded their absence as a proof of extinction. Anyway, detection of adults should be performed to confirm the extinction. Are there some data about this? Finally, there are some typos: line 308, correct "join teffect"; line 382-83, format character size. In conclusion, the paper canbe accepted with minor revisions.

Author Response

The authors present the results of a long-term monitoring on the vulnerable species Rana latastei in the Monza Park and document the lack of clutches in the last years, concluding that the species went extinct in the study area. The paper is well written and the data are clearly presented to support the conclusions. There are some points that should be better explained: 1) being the frog vulnerable, authors should have permissions from local authorities to study the species and cite them in the text; 2) for the same reasons, details about field procedures and mitigating measures should be presented, e.g., disinfection of the equipment such as boots or gloves 3) as an example of recent extinction of anurans authors cite in the introduction the Costarican golden toad Incilius periglenes. Among the possible causes of the extinction the pathogenic chytrid fungus Batrachochytrium dendrobatidis has been hypothesized. Did the authors consider the fungus as a possible cause of disappearence R. latastei? 4) The authors monitored the presence of clutches as an estimator of decline and regarded their absence as a proof of extinction. Anyway, detection of adults should be performed to confirm the extinction. Are there some data about this? Finally, there are some typos: line 308, correct "join teffect"; line 382-83, format character size. In conclusion, the paper canbe accepted with minor revisions.

RESPONSE: We thank the reviewer for the positive and constructive comments. We reply point by point below:

  • There was no manipulation of individuals for this study, we only counted the number of clutches and recorded adults/larvae spotted during visual encounter surveys. Since direct manipulation of animals was not necessary, we think that a permission number is not required in this case. In any case, we are available to provide permission number for animal manipulation if the editor thinks it is the case.
  • We added the description of the disinfection protocol used to avoid the transmission of pathogens (lines 136-137).
  • Unfortunately, data on the presence of dendrobatidis in the area are not available. Some projects are starting with the aim to expand the knowledge about the occurrence of this pathogen in northern Italy, but these are still at early stages.
  • No adults, calls, eggs, or larvae were detected in 2022 and 2023. We added this information into the discussion. Additionally, we pointed out in the discussion that in principle the species might be still present, given that we may have missed some adults or a very small portion of egg clutches (lines 372-382).

We also corrected the mentioned typos, thank you for noticing.

Round 2

Reviewer 2 Report

Overall much improved, thanks! And with the water management and crayfish interest of the paper increases somewhat so well done. All the edits were good the only one is dealt below and I offer just a minor adjustment. And there were a few italics issues with species names which I am sure will be picked up during type setting.

lines 125-127: During a survey, each observer walked a section of the wetland, hence each portion of the wetland is explored only once during a single survey. This allows to optimize time when more observers are available, without influencing detections.

Try: During a survey, each observer walked a section of the wetland, hence each portion of the wetland was explored only once during a single survey. This approach was taken to optimize time when more observers are available, without influencing detections.

Much improved and it is clear the authors did a good job of checking over their work again

Author Response

we modified the manuscript following this suggestion